# A Week of Sleep Restriction Does Not Affect Nighttime Glucose Concentration in Healthy Adult Males When Slow-Wave Sleep Is Maintained

**DOI:** 10.3390/s22186962

**Published:** 2022-09-14

**Authors:** Thomas G. Kontou, Charli Sargent, Gregory D. Roach

**Affiliations:** Appleton Institute for Behavioural Science, Central Queensland University, Wayville, SA 5034, Australia

**Keywords:** continuous glucose monitoring, polysomnography, sleep architecture, metabolism, sedentary activity, wrist actigraphy, rapid eye movement sleep, insulin, diet, glucose tolerance

## Abstract

The aim of this laboratory-based study was to examine the effect of sleep restriction on glucose regulation during nighttime sleep. Healthy males were randomly assigned to one of two conditions: 9 h in bed (*n* = 23, age = 24.0 year) or 5 h in bed (*n* = 18, age = 21.9 year). Participants had a baseline night with 9 h in bed (23:00–08:00 h), then seven nights of 9 h (23:00–08:00 h) or 5 h (03:00–08:00 h) in bed. Participants were mostly seated during the daytime but had three bouts of treadmill walking (4 km·h^−1^ for 10 min) at ~14:40 h, ~17:40 h, and ~20:40 h each day. On the baseline night and night seven, glucose concentration in interstitial fluid was assessed by using continuous glucose monitors, and sleep was assessed by using polysomnography. On night seven, compared to the 9 h group, the 5 h group obtained less total sleep (292 min vs. 465 min) and less REM sleep (81 min vs. 118 min), but their slow-wave sleep did not differ (119 min vs. 120 min), and their glucose concentration during sleep did not differ (5.1 mmol·L^−1^ vs. 5.1 mmol·L^−1^). These data indicate that sleep restriction does not cause elevated levels of circulating glucose during nighttime sleep when slow-wave sleep is maintained. In the future, it will be important to determine whether increased insulin is required to maintain circulating glucose at a normal level when sleep is restricted.

## 1. Introduction

Glucose concentration in the blood is tightly controlled within a normal range (i.e., euglycemia). Euglycemia is achieved through complex metabolic pathways, predominantly influenced by the production of insulin [1]. Consuming a high-sugar diet or being overweight are examples of two risk factors that can result in irreversible damage to the metabolic system and potentially lead to the development of type 2 diabetes [2,3]. Another risk factor for impaired glucose metabolism and the development of type 2 diabetes is sleep restriction [4,5,6]. The links between sleep restriction and type 2 diabetes have been identified with epidemiological research [7] and the potential mechanisms for this relationship have been established with experimental research [8,9]. However, one important aspect of this relationship has not yet been established, which is whether glucose metabolism may become impaired by sleep restriction while the subject is asleep.

Under tightly controlled laboratory conditions, consecutive nights of sleep restriction can cause impairment to the metabolic system. This damage can manifest as impaired glucose tolerance [10] and insulin resistance [11,12] after fewer than six hours of time in bed per night. The responsiveness of the metabolic system to multiple nights of sleep restriction in these studies is typically assessed by using glucose-tolerance tests (for e.g., [13]). The tests are conducted when participants are in a fasting state after waking and involve a glucose bolus—administered either orally or intravenously—followed by frequent blood sampling to assess glucose and insulin concentrations. This approach is useful because it allows for the observation of the performance of the metabolic system when it is challenged under standardized conditions during the waking period. However, this method does not capture whether glucose concentrations are altered by sleep restriction during sleep. Understanding the influence of different behaviors on fluctuations of glucose concentrations during sleep is important [14,15]. This is because, under ideal conditions, humans may spend approximately one-third of their lives asleep [16]. Therefore, impaired glucose tolerance during sleep may have a cumulative detrimental effect on the metabolic system and significantly contribute to the overall risk of developing type 2 diabetes.

Changes in sleep architecture induced by sleep restriction may influence glucose concentration during sleep. The duration of stage one, stage two, and REM sleep typically decrease with multiple nights of sleep restriction, whereas the duration of slow-wave sleep (SWS) remains relatively unchanged [17,18]. These changes in sleep architecture may influence glucose concentration during nighttime sleep. This is because glucose concentrations are usually lower during REM sleep and higher during SWS [19,20,21,22]. The link between sleep architecture and glucose metabolism has been demonstrated in individuals with sleep apnea. Continuous positive airway pressure (CPAP) treatment in patients with sleep apnea increased the duration of REM sleep, reduced the number of nighttime arousals, and lowered mean glucose levels during an entire nighttime sleep period [23]. Therefore, normalizing sleep architecture had an important effect on improving glucose concentrations during sleep. However, this does not indicate how sleep restriction may influence glucose concentration during sleep in otherwise healthy individuals.

Sleep restriction among healthy individuals may alter glucose metabolism during sleep. For example, when sleep is restricted to 4.5 h in bed per night for three nights, participants experienced insulin resistance during the nighttime (i.e., higher area under the curve for insulin in the sleep-restriction condition compared with the control condition) but no change in glucose concentration (5.08 ± 0.07 mmol·L^−1^ during sleep restriction vs. 5.05 ± 0.07 mmol·L^−1^ during the control condition) [24]. However, glucose concentrations were measured between 21:30 h and 09:00 h in both conditions, whereas participants were allocated time in bed from 23:00 h to 07:30 h in the control condition and from 01:00 h to 05:30 h in the sleep-restriction condition [24]. Therefore, the potential impact of sleep architecture on glucose concentration may not have been captured. With more severe sleep restriction (4 h time in bed) for a longer period (five nights), average glucose concentration remained elevated over an entire 24 h period (including the nighttime) after the fifth night of sleep restriction [25]. However, in this study, separate analyses of glucose concentrations during the sleep period were not conducted.

To determine the influence of sleep restriction on glucose concentrations during a sleep period, separate analyses of the sleep period would be required, as well as an assessment of the potential influence of sleep stage on glucose concentrations. This would significantly contribute to the literature in this field because it is not known if, and to what extent, glucose metabolism may become impaired during sleep when practicing sleep restriction. Any impairment to glucose metabolism during sleep, if maintained for months or years, may have cumulative detrimental effects on the metabolic system and contribute to the risk of developing type 2 diabetes. Thus, the aims of the present study were to (i) examine the impact of consecutive nights of moderate sleep restriction (i.e., 5 h spent in bed) on interstitial glucose concentrations during a sleep period compared to a control condition (i.e., 9 h spent in bed) in healthy males; and (ii) compare glucose concentrations during the N2, N3, and REM stages of sleep between the moderate sleep-restriction condition and the control condition. It was hypothesized that (i) restricting time in bed to five hours per night will decrease the proportion of REM sleep and increase the proportion of slow-wave sleep; and (ii) glucose concentration will be higher in the moderate sleep-restriction condition compared with the control condition.

## 2. Materials and Methods

### 2.1. Participants

Forty-eight healthy males volunteered to participate in the study and obtained either 9 h (*n* = 26) or 5 h (*n* = 22) time in bed during the experimental phase of the study (Figure 1). Participants were non-smokers, not on a diet, had not experienced any weight gain or loss in the three months prior to participation, did not drink excessive amounts of alcohol or caffeine, had normal fasting glucose, and were not suffering from any metabolic or sleep disorders. Participants had not undertaken any shiftwork or transmeridian travel in the three months prior to the study’s commencement. Interested participants completed a screening health questionnaire and were invited to the sleep laboratory for a familiarization session and to provide a fasting capillary blood sample for glucose measurement. The project was approved by CQ University’s Human Research Ethics Committee, and participants provided written informed consent prior to study commencement.

In the 9 h condition, data were not available for one participant due to equipment malfunction, and one participant obtained 4.9 h of sleep (on BL), which was (i) equal to the mean total sleep time in the 5 h condition and (ii) two standard deviations outside the mean of total sleep time during BL for the 9 h condition. Data from this participant were excluded from the analyses. Another participant in the 9 h condition obtained 5.1 h of sleep on BL, and this participant was also excluded from the analyses. In the 5 h condition, three participants withdrew from the study, and data were not available for one participant due to equipment malfunction. Data were available for 23 participants in the 9 h condition, with a mean (±SD) age and body mass index of 24.0 ± 3.5 years and 23.2 ± 1.7 kg·m^−2^, respectively. Data were available for 18 participants who completed the 5 h condition with a mean (±SD) age and body mass index of 21.9 ± 4.2 years and 23.3 ± 1.8 kg·m^−2^, respectively. The mean profiles of activity in the 9 h and 5 h conditions during BL and E7 are presented in Figure 2g,h.

### 2.2. Continuous Glucose Monitoring Device

Interstitial glucose concentrations were assessed overnight, using a continuous glucose-monitoring device (Medtronic Guardian; Medtronic, Northridge, CA, USA). The system consists of a sensor, a transmitter, and a recording device. The sensor (Enlite Glucose Sensor; Medtronic, Northridge, CA, USA) is a thin (31-gauge), short (8.5-mm) substrate with electrode surfaces to detect glucose concentrations. The transmitter is a small oval-shaped device approximately 2 cm across and is attached to the exposed part of the sensor and covered with a waterproof dressing. The transmitter sends radio signals to the recording device, and interstitial glucose concentrations are recorded in 5 min epochs. The devices were chosen because they are minimally invasive and, once inserted, allow for continuous monitoring of glucose concentrations without disturbing the wearer, as is essential during sleep.

### 2.3. Capillary Blood Sampling Device

Capillary blood samples were collected to measure blood glucose concentrations to calibrate the continuous glucose-monitoring device. A 28-gauge lancet with a penetration depth of 1.6 mm (Haemolance Plus^®^ Micro Flow; Ozorkow, Poland) was used to obtain each sample. Capillary blood samples were analyzed with a portable glucose meter (Accu-Chek^®^ Performa; Manaheim, Baden-Württemberg, Germany) in combination with glucose test strips (Accu-Chek^®^ Performa test strips; Manaheim, Baden-Württemberg, Germany).

### 2.4. Sleep Monitoring

Sleep was assessed by using polysomnography (PSG; Compumedics Grael; Melbourne, VIC, Australia). The montage of electrodes included two electroencephalograms, two electrooculograms, and two electromyograms. Sleep records were analyzed in 30 s epochs by a single technician, using established criteria [26]. Total sleep time (TST); wake after sleep onset (WASO); and the duration of time spent in the N1, N2, N3, and REM sleep stages were calculated for each record.

### 2.5. Activity Monitoring

Sedentary activity was assessed by using activity monitors (Actical Z-series; Philips Respironics; Bend, OR, USA). The activity monitors were configured to sum and store data in 30 s epochs based on activity counts from a piezoelectric accelerometer with a sensitivity of 0.05 g and a sampling rate of 32 Hz [27]. The activity monitors were firmly attached to the inside of a waist bag that was secured to the hip of each participant. This position is associated with high sensitivity and specificity for measuring sedentary, moderate, and vigorous physical activity [28].

### 2.6. Laboratory Setting

Participants lived in the sleep laboratory at the Appleton Institute for Behavioral Science for the duration of the study. The laboratory contains six private bedrooms, each with an adjoining living space and bathroom, and a communal dining area. The ambient temperature in the laboratory was maintained between 21 and 23 °C. The light levels were maintained at 350 lux during wake periods and were extinguished during sleep periods. In the sleep-restriction condition, the light levels were reduced to <10 lux between 23:00 h and 03:00 h to minimize potential phase delays in the circadian timing system [29].

### 2.7. Experimental Design

Participants lived in the sleep laboratory 24 h per day for 11 consecutive days (Figure 1). Participants were randomly assigned to a control condition (i.e., 9 h time in bed) or a moderate sleep-restriction condition (i.e., 5 h time in bed). Time in bed was experimentally manipulated by delaying participants’ bedtime in the sleep-restriction condition. The experimental design allowed for comparisons to be made both within (i.e., pre-post) and between groups (i.e., 9 h condition vs. 5 h condition). The protocol began with an adaptation sleep (23:00–08:00 h) to familiarize participants with the equipment used to monitor sleep, and a baseline sleep (23:00–08:00 h) to establish baseline characteristics of sleep. For the next 7 consecutive nights (i.e., E1–E7), participants spent either 9 h in bed, from 23:00–08:00 h (*n* = 26), or 5 h in bed, from 23:00–08:00 h (*n* = 22). The average glucose concentration on E7 was chosen for the analyses because, on E7, participants experienced the maximum amount of cumulative sleep restriction in the study, and seven nights of sleep restriction was longer than previous investigations where glucose concentrations were measured at night, during restricted sleep (four [24] or five nights [25]). Participants were only allowed to sleep during the allocated bedtimes—compliance was ensured by researchers, who monitored participants in person and via closed-circuit television displaying the participants’ living areas. The only physical activity that participants were permitted to engage in was three 10-minute walking sessions per day on a motorized treadmill, at a speed of 4 km·h^−1^ at ~14:40 h, ~17:40 h, and ~20:40 h.

Glucose sensors were inserted on the adaptation day (AD) between 13:00 h and 14:00 h. The sensors were inserted approximately 5 cm below the navel either to the left or right. Following insertion, the sensors entered a 2 h warm-up period, and then the sensors were calibrated by using a capillary blood sample. The sensor inserted on AD was ‘restarted’ at 11:00 h on experimental day 1 (E1, after wearing the sensor for 72 h), as per the manufacturer’s instructions. At 16:00 h on experimental day 3 (E3), sensors were removed, and new sensors were inserted. These sensors were ‘restarted’ at approximately 15:00 h on experimental day 6 (E6). The devices were calibrated three times per day, at 08:30 h, 15:30 h, and 22:30 h, according to the manufacturer’s guidelines (Figure 1).

**Figure 1 sensors-22-06962-f001:**
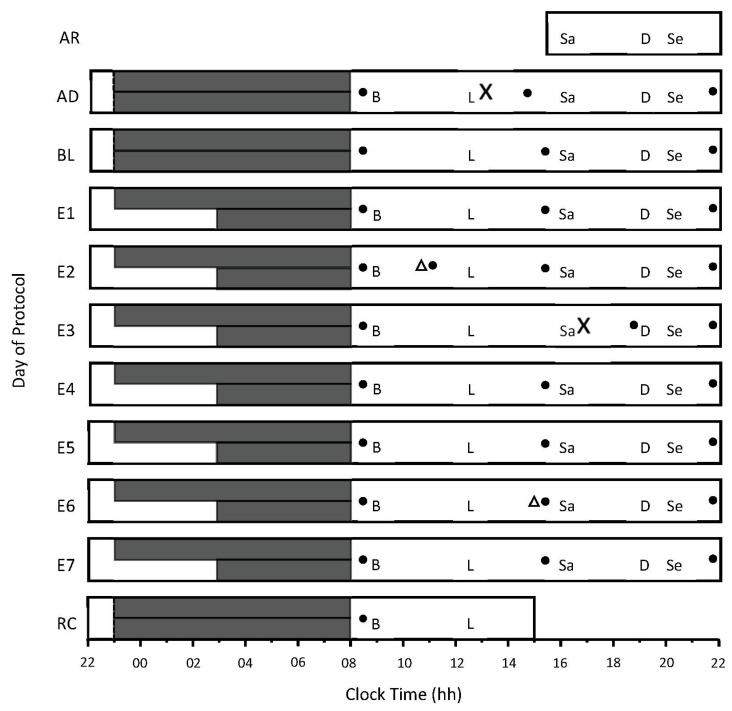
Study protocol. The *x*-axis indicates time of day, from 22:00 h to 22:00 h. The *y*-axis indicates the day of the protocol, including AR = arrival day, AD = adaptation day, and BL = baseline day. E1 to E7 = experimental days 1 to 7, and RC = recovery day. The long and short solid dark gray bars represent time in bed for the 9 h and the 5 h conditions, respectively. Meals and snacks were served at the same time each day—B = breakfast, L = lunch, Sa = afternoon snack, and Se = evening snack. ‘X’ indicates when the first and second continuous glucose-monitoring sensors were inserted, and the solid black circles represent capillary-blood-testing times for device calibration. Triangles represent when the sensors were restarted to ensure their continued function beyond 72 h of wear.

### 2.8. Meals and Caloric Intake

Participants were provided with a eucaloric diet during the protocol. Kilojoule calculations were based on the Harris–Benedict Equation [30], using an average activity factor of 1.4, which reflects sedentary (1.2) to moderately active (1.5) activity [31,32]. Participants were provided with three main meals per day (breakfast, lunch, and dinner) and afternoon and evening snacks (Figure 1). These meals provided approximately 7530 kJ per day, per participant (Table 1). To tailor the kilojoule intake for each participant’s age, height, and weight, additional kilojoules were provided after lunch, afternoon snack, and dinner. These kilojoules were provided in the form of savory (unsalted) crackers, and equal quantities (by weight) of mixed salted nuts, cheese, and cold processed meat.

### 2.9. Data Analysis

Sleep and glucose concentrations recorded on baseline night and experimental day seven (E7) were included in the analyses. Sleep and glucose concentrations were recorded in 30 s and 5 min epochs, respectively. Due to the difference in epoch length, sleep stages and glucose concentrations were aligned manually. To achieve this manual alignment, the glucose concentration obtained for a 5 min epoch was assigned to each of the 10 × 30 s epochs at the corresponding clock time in that 5 min epoch. The glucose concentration was included in the analyses only if every 30 s epoch of sleep that occurred during the 5 min epoch for the glucose concentration was the same sleep stage. Once the glucose data were aligned with the sleep-stage data, a 15 min delay was applied to the glucose data. This is because interstitial glucose concentrations appear up to 15 min later than they do in blood [33]. Sleep stages N2, N3, REM, and WASO were included in the analyses. Sleep stage N1 was excluded from the analyses, as there were too few 5 min epochs where every epoch was stage N1. Average glucose concentration was chosen because the sleep lengths in the control condition (9 h time in bed) and the experimental condition (5 h time in bed) were vastly different and, therefore, other calculations (e.g., area under the curve) would not have allowed for valid comparisons.

### 2.10. Statistical Analysis

The normality of all variables was determined by using the Shapiro–Wilk test. To compare total sleep time and sleep architecture between the 9 h and 5 h conditions on days BL and E7, separate one-way factorial ANOVAs were conducted. In each analysis, the independent variable was time in bed (9 h and 5 h), and the dependent variables were total sleep time; wake after sleep onset; and minutes of the N2, N3, and REM sleep stages.

The effects of the time in bed (and study day) on glucose concentration were examined by using a two-way mixed ANOVA. The time in bed (2 levels—9 h and 5 h) was included as a between-subjects factor, and the study day (2 levels—BL and E7) was included as a within-subjects factor. The average glucose concentration (mmol·L^−1^) during the time in bed was the dependent variable. For the BL sleep, the average glucose concentration was calculated between 23:00 and 08:00 h in both conditions; for the E7 sleep, the average glucose concentration was calculated between 23:00 and 08:00 h in the 9 h condition and between 03:00 and 08:00 h in the 5 h condition. The ANOVA was used to test for main effects of the independent variables and for an interaction effect between the time in bed and study day.

The effects of sleep stage (and time in bed) on glucose concentration were examined by using a two-way mixed ANOVA. The sleep stage during the time in bed (4 levels—wake, stage 2, stage 3, and REM sleep) was included as a within-subjects factor, and the time in bed (2 levels—9 h and 5 h) was included as a between-subjects factor. Separate ANOVAs were conducted for BL and E7. The average glucose concentration (mmol·L^−1^) during a particular sleep stage was the dependent variable. The ANOVA was used to test for the main effects of the independent variables and for an interaction effect between the sleep stage and time in bed.

Sphericity was examined in all ANOVAs, using Mauchly’s test of sphericity. If the assumption of sphericity was violated, *p*-values were based on Greenhouse–Geisser’s corrected degrees of freedom, but the original degrees of freedom are reported. Where appropriate, pairwise comparisons were performed with a Bonferroni correction for multiple comparisons. All analyses were conducted by using SPSS version 28 (IBM Corp, Armonk, NY, USA) and were considered significant at *p* < 0.05. All data are reported as mean ± SD.

## 3. Results

### 3.1. Sleep

There was no difference in total sleep time, wake after sleep onset, or time spent in any stage of sleep between the 9 h and 5 h conditions at BL (Table 1). On E7, participants in the 5 h condition obtained less sleep (-174 min), spent less time in stage N2 (-113 min) and stage REM (-37 min), and spent less time awake after sleep onset (-43 min) compared with participants in the 9 h condition (Table 2). There was no difference in the time spent in stage N3 sleep between conditions.

### 3.2. The Effect of Time in Bed on Average Glucose Concentration during Sleep

The average glucose concentration during the time in bed on the baseline night was 4.4 ± 0.1 mmol·L^−1^ in the 9 h condition and 4.6 ± 0.1 mmol·L^−1^ in the 5 h condition (Figure 2). The average glucose concentration during the time in bed on experimental night seven was 5.1 ± 0.1 mmol·L^−1^ in the 9 h condition and 5.1 ± 0.1 mmol·L^−1^ in the 5 h condition (Figure 2). There was no main effect of the time in bed on average glucose concentration during the time in bed (F_1,39_ = 0.02; *p* = 0.896), and there was no interaction between the time in bed and study day on the average glucose concentration (F_1,39_ = 0.74, *p* = 0.395), but there was a main effect of the study day on average glucose concentration (F_1,39_ = 18.6, *p* < 0.001). The average glucose concentration was lower during the BL sleep period than during the E7 sleep period (4.5 ± 0.7 mmol·L^−1^ vs. 5.2 ± 0.6 mmol·L^−1^; *p* < 0.001). All glucose concentrations were similar between conditions, with the largest difference between time-in-bed conditions being 0.4 mmol·L^−1^, which occurred during WASO on E7, (9 h vs. 5 h; see Table 3).

**Table 3 sensors-22-06962-t003:** Average glucose concentrations during baseline night and experimental night E7 in the 9 h and 5 h conditions.

Variable	Condition	
	9 h	5 h
*Baseline*		
Glucose concentration during time in bed (mmol·L^−1^)	4.4 ± 0.1	4.6 ± 0.1
Glucose concentration during stage N2 (mmol·L^−1^)	4.4 ± 0.9	4.5 ± 0.6
Glucose concentration during stage N3 (mmol·L^−1^)	4.5 ± 0.8	4.5 ± 0.6
Glucose concentration during stage REM (mmol·L^−1^)	4.6 ± 0.9	4.6 ± 0.6
Glucose concentration during WASO (mmol·L^−1^)	4.7 ± 0.6	4.7 ± 0.6
*Experimental Day 7*		
Glucose concentration during time in bed (mmol·L^−1^)	5.1 ± 0.1	5.1 ± 0.1
Glucose concentration during stage N2 (mmol·L^−1^)	5.3 ± 0.5	5.1 ± 0.7
Glucose concentration during stage N3 (mmol·L^−1^)	5.2 ± 0.6	5.1 ± 0.6
Glucose concentration during stage REM (mmol·L^−1^)	5.3 ± 0.5	5.1 ± 0.7
Glucose concentration during WASO (mmol·L^−1^)	5.5 ± 0.7	5.1 ± 0.5

Data are mean ± SD. REM, rapid eye movement; WASO, wake after sleep.

**Figure 2 sensors-22-06962-f002:**
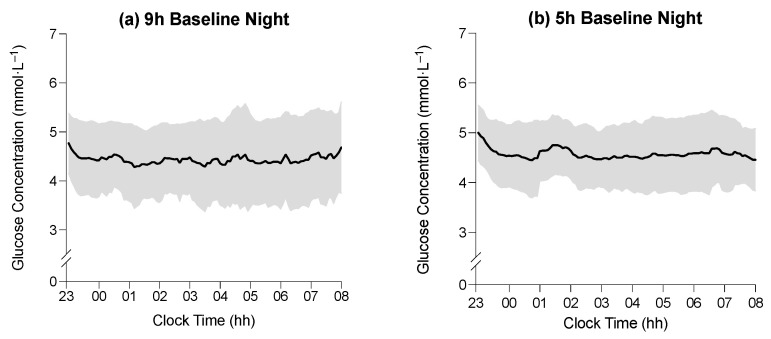
(**a**–**d**) Average glucose concentration (black line) in the 9 h condition at baseline, the 5 h condition at baseline, the 9 h condition on experimental night seven, and the 5 h condition on experimental night seven, respectively. (**e**,**f**) Difference in average glucose concentration between baseline and experimental night seven in the 9 h and 5 h conditions, respectively. Horizontal axes indicate clock time (hh = hours), and vertical axes represent glucose concentrations (mmol·L^−1^). Shaded regions indicate ± SD. (**g**,**h**) Mean activity profiles in the 9 h condition during baseline, the 5 h condition during baseline, the 9 h condition during experimental day/night seven, and the 5 h condition during experimental day/night seven, respectively (**i**,**j**). Horizontal axes indicate clock time (hh = hours), and vertical axes represent mean activity counts (‘000).

### 3.3. The Effect of Time in Bed on Average Glucose Concentration during Sleep Stages

For the BL sleep period, there was no main effect of time in bed on average glucose concentration during wake after sleep onset and the N2, N3, and REM (F_1,33_ = 1.67, *p* = 0.205) sleep stages and no interaction between the time spent in bed and sleep stage for average glucose concentration during wake after sleep onset and the N2, N3, and REM (F_3,99_ = 0.87, *p* = 0.828) sleep stages; however, there was a main effect of sleep stage on average glucose concentration (F_3,99_ = 4.43, *p* = 0.033). The average glucose concentration appeared to be the highest during periods of wake after sleep onset (4.8 ± 0.6 mmol·L^−1^) and lowest during stage N2 (4.5 ± 0.7 mmol·L^−1^), but pairwise comparisons between sleep stages were not significant (Figure 3).

For the E7 sleep period, there was no main effect of time in bed on average glucose concentration during wake after sleep onset and the N2, N3, and REM (F_1,30_ = 2.29, *p* = 0.141) stages and no interaction between the time spent in bed and sleep stage for average glucose concentration during wake after sleep onset and the N2, N3, and REM (F_3,90_ = 0.54, *p* = 0.545) stages of sleep; however, there was a main effect of sleep stage on the average glucose concentration (F_3,90_ = 4.58, *p* = 0.022). The average glucose concentration appeared to be highest during periods of wake after sleep onset (5.3 ± 0.6 mmol·L^−1^) and lowest during the REM stage (5.1 ± 0.6 mmol·L^−1^), but pairwise comparisons between sleep stages were not significant (Figure 3).

## 4. Discussion

The aims of the present study were to (i) examine the impact of consecutive nights of moderate sleep restriction (i.e., 5 h time in bed) on interstitial glucose concentration during sleep and (ii) determine if moderate sleep restriction influenced glucose concentration in a manner dependent on stage of sleep. The main findings were that (i) there was no effect of moderate sleep restriction on average glucose concentration during sleep, and (ii) while there was an effect of sleep stage on average glucose concentration, the difference in glucose concentration between sleep stages appeared minimal.

In the present study, moderate sleep restriction did not influence glucose concentrations during sleep periods. Multiple nights of sleep restriction can influence glucose metabolism during wake periods such that (i) the average glucose concentration is elevated throughout a wake period [25], and (ii) glucose metabolism (tested during a glucose challenge) is impaired [10]. Glucose concentration in the present study was measured during sleep—a period which is characterized by the post-absorptive state (i.e., ~6 h after meal ingestion) and the fasting state (i.e., ~10 h after meal consumption) [34]. Therefore, in healthy individuals, fasting glucose concentrations are maintained during restricted sleep. Indeed, following multiple nights of sleep restriction, a single fasting-glucose sample taken upon awakening is also usually in the normal range (i.e., <5.5 mmol·L^−1^) [35,36,37,38]. There was a small non-significant difference (0.2 mmol·L^−1^) at baseline between the 9 h and the 5 h time-in-bed conditions. This difference was unlikely to represent a clinical difference between groups (9 h and 5 h) because the average glucose concentration was well within the healthy fasting range (<5.5 mmol·L^−1^; 4.4 ± 0.1 mmol·L^−1^ in the 9 h condition and 4.6 ± 0.1 mmol·L^−1^ in the 5 h condition) [39].

In contrast to the findings in the present study, there is some evidence to indicate that glucose metabolism during sleep may be affected by sleep restriction. For example, when time in bed is restricted to below 5 h per night, glucose concentration is elevated during sleep [25], and insulin resistance may also occur [24]. In the present study, the magnitude of sleep restriction was less (i.e., 5 h time in bed per night), and the slow-wave sleep duration was preserved in the sleep-restriction condition (i.e., the duration of stage N3 sleep on experimental night seven in the 5 h condition was similar to that observed at baseline and on E7 in the 9 h condition). The preservation of slow-wave sleep during the sleep period may be an important protective mechanism against the metabolic impact of sleep restriction. For example, when sleep is experimentally fragmented to reduce the total minutes of slow-wave sleep while maintaining normal total sleep time, insulin sensitivity decreases, and glucose tolerance is reduced [40,41]. Conversely, when sleep is experimentally fragmented to reduce total time in REM sleep while maintaining normal total sleep time and slow-wave sleep duration, glucose tolerance is not impaired [42].

No impairment to the metabolic system during sleep was identified because of sleep restriction. However, only one aspect of the metabolic system was observed (glucose concentration). Insulin, for example, is an important hormone that is produced in the pancreas to help maintain glucose concentrations at the fasting level. It is possible that participants experienced insulin resistance and produced more insulin to maintain glucose concentration at fasting levels, with no manifest impairment to glucose concentration [12,24,43]. However, the assessment of insulin concentration requires invasive techniques (e.g., indwelling catheter) compared with continuous glucose-monitoring devices and may be disruptive to sleep.

The average glucose concentration during the final sleep period (i.e., experimental day 7) was higher (+0.7 mmol·L^−1^) compared with average glucose concentrations during the baseline sleep period. One potential explanation for this increase is a reduction in physical activity during the protocol. Participants in both time-in-bed conditions completed identical bouts of physical activity each day (i.e., three ×10 min bouts of walking at 4 km·h^−1^ on a motorized treadmill). This pace of walking is equivalent to 3.3 metabolic equivalents and is consistent with the American College of Sports Medicine’s daily physical activity recommendations for maintaining health [44]. However, this level of physical activity may not be sufficient to maintain healthy glycemic control. For example, three days of reduced physical activity (i.e., 10,000 steps per day reduced to 5000 steps per day) in healthy, free-living individuals increases post-prandial glucose concentration by ~0.4 mmol·L^−1^ [45]. The results of the present study suggest that daily physical activity may be an important mediator of glycemic control [45,46].

There are some methodological issues to consider when interpreting the results of the present study. Participants were provided a eucaloric diet throughout the protocol. This approach negates the potential confounding effect of an uncontrolled diet (i.e., ad libitum food consumption) on glucose metabolism. Ad libitum food intake, in combination with sleep restriction, can result in the consumption of calorie-dense, nutrient-poor foods and weight gain [8,37]. Weight gain is a risk factor for the impairment of glucose metabolism via insulin resistance [47], and for this reason, food intake in the present study was controlled. Nevertheless, a eucaloric diet may lack ecological validity. In the sleep-restriction condition, participants obtained an average of 4.9 h of sleep per night. This may be considered severe sleep restriction, but up to 30% of the working population reports sleeping 5 h or less [48]. Severe sleep restriction (e.g., 4 h time in bed per night) may impair glucose metabolism during sleep [24], but obtaining 4 h of sleep per night is unlikely to be sustained for long periods of time because, with that amount of time in bed, maintaining wakefulness is difficult [49].

## 5. Conclusions

Taken together, the results of the present study indicate that seven consecutive nights of moderate sleep restriction does not affect glucose concentration during sleep in healthy adult males. In the future, it will be important to determine whether other hormones involved in glucose metabolism (e.g., insulin, cortisol, norepinephrine, and epinephrine) are impaired during sleep periods when the time in bed is restricted.

## Figures and Tables

**Figure 3 sensors-22-06962-f003:**
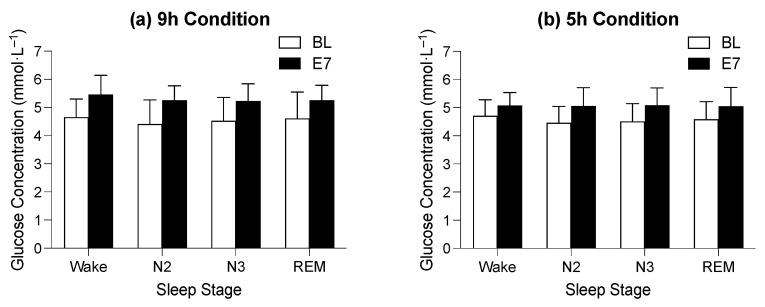
Figure indicates the difference between average glucose concentration on baseline (BL, white bars) and experimental night seven (E7, black bars) in the 9 h time-in-bed condition (**a**) and the 5 h time-in-bed condition (**b**) in the different sleep stages. The *y*-axis indicates glucose concentration in mmol·L^−1^, and the *x*-axis indicates sleep stages where Wake = wake after sleep onset, N2 = stage N2, N3 = stage N3, and R = REM sleep.

**Table 1 sensors-22-06962-t001:** Components of meals and snacks consumed during the study protocol.

Variable	Condition	
	9 h	5 h
*Breakfast*		
Protein (%)	17 ± 2	17 ± 3
Fat (%)	25 ± 3	26 ± 4
Carbohydrate (%)	59 ± 6	58 ± 5
Energy (kJ)	2309 ± 265	2301 ± 284
*Lunch*		
Protein (%)	23 ± 5	22 ± 5
Fat (%)	40 ± 13	43 ± 13
Carbohydrate (%)	37 ± 6	35 ± 8
Energy (kJ)	2545 ± 623	2848 ± 733
*Afternoon Snack*		
Protein (%)	18 ± 6	19 ± 6
Fat (%)	48 ± 13	47 ± 14
Carbohydrate (%)	33 ± 7	35 ± 7
Energy (kJ)	2371 ± 624	2616 ± 691
*Dinner*		
Protein (%)	25 ± 17	24 ± 4
Fat (%)	40 ± 29	40 ± 13
Carbohydrate (%)	36 ± 6	36 ± 4
Energy (kJ)	2992 ± 644	3183 ± 662
*Evening Snack*		
Protein (%)	13 ± 3	12 ± 2
Fat (%)	32 ± 12	31 ± 15
Carbohydrate (%)	56 ± 19	56 ± 15
Energy (kJ)	1056 ± 358	1017 ± 322

Data are mean ± SD.

**Table 2 sensors-22-06962-t002:** Sleep characteristics at baseline and on experimental day E7 in the 9 h and 5 h conditions.

Variable	Condition		F	*p*-Value
	9 h	5 h		
*Baseline*				
TST (min)	477.7 ± 40.0	490.6 ± 22.5	1.45	0.236
Stage N2 (min)	199.8 ± 28.2	217.0 ± 37.5	2.53	0.120
Stage N3 (min)	124.0 ± 33.8	118.6 ± 35.0	0.23	0.639
Stage REM (min)	121.33 ± 25.5	126.7 ± 24.3	0.43	0.567
WASO (min)	41.1 ± 26.3	32.9 ± 20.9	1.10	0.323
*Experimental Day 7*				
TST (min)	465.7 ± 40.9	292.1 ± 4.9	336.8	<0.001
Stage N2 (min)	196.3 ± 36.5	83.7 ± 19.5	148.1	<0.001
Stage N3 (min)	120.2 ± 45.9	119.3 ± 25.0	0.01	0.940
Stage REM (min)	118.4 ± 20.7	81.1 ± 16.3	42.0	<0.001
WASO (min)	48.7 ± 30.5	6.1 ± 4.5	36.3	<0.001

Data are mean ± SD. TST, total sleep time; REM, rapid eye movement; WASO, wake after sleep onset.

## Data Availability

The datasets generated from the study are available from the corresponding author on reasonable request.

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
