# Peer review of "A Week of Sleep Restriction Does Not Affect Nighttime Glucose Concentration in Healthy Adult Males When Slow-Wave Sleep Is Maintained"

_sensors, 2022, doi:10.3390/s22186962_

Round 1

Reviewer 1 Report

1)      The topic of this paper discusses a study of effect of sleep restriction on glucose regulation during night-time sleep. The work is interesting and but the paper fails to reflect the novelty of the work. There is a very minimal novel contribution at the system level presented. The paper is preliminary report and need further detailed analysis and experimentations.

2)      The introduction section is written too much weak and has no new information is provided. Need complete rewriting with some up-to-date references.3)      It would be better if a comparison with state-of-the-art works is added. Otherwise, the manuscript may be read more like a project report. The current manuscript doesn't have any comparison table, moreover, the experiment design choices should be elaborated along with a state-of-the-art quantitative/qualitative comparison.4) Paper flow needs to be revised and sections are not much descriptive. It becomes difficult to follow the paper.

Author Response

Please see the attachment:

Reviewer 2 Report

This paper studied the night-time glucose concentration in healthy male adults whose sleep was restricted within a week. The results showed that reduced sleep duration did not lead to higher glucose concentrations during sleep when the length of deep sleep was constant. This is an interesting study. In addition, the paper was well organized. However, the paper needs to be revised.

1.     The title is “A week of sleep restriction does not affect night-time glucose concentration in healthy young adults when slow-wave sleep is maintained”, which is inconsistent with the content of the study. The subjects were 48 healthy males in the experiment. Whether gender has an effect on glucose concentration during sleep restriction? I suggest change the title.

2.     Section 3.1 and 3.2 were the characteristics of participants and sleep time. These contents should be moved to Section 2 Materials and Methods.

3.     Line283-284: “The average glucose concentration during time in bed on the baseline night was 4.4 ± 0.1 mmol·l-1 in the 9-h condition and 4.6 ± 0.1 mmol·l-1 in the 5-h condition”. There is a significant difference between 9-h condition and 5-h condition. However, the difference disappeared after a week experiment. The phenomenon is strange. In other words, the subjects are fundamentally difference in the 9-h condition at baseline and in the 5-h condition at baseline.

4.     Line 228: “3.0 Statistical Analysis”. It should be 2.10…

5.     What is the index “F” mean in table 2? And what is the subscript of F mean? For example F1,39 in Line 288 and F3,99 in Line 313.

6.     Please cite more references from the last 3 years.

Author Response

Please see the attachment:

Round 2

Reviewer 1 Report

Thank you. My comments are addressed.